neuroscience/psychology

audience effects, joint action, action observation, EEG, mu-rhythm

**Author for correspondence:**
Marius Zimmermann
e-mail: marius.zimmermann@ur.de

# Intra-individual behavioural and neural signatures of audience effects and interactions in a mirror-game paradigm

Marius Zimmermann[1,2], Arianna Schiano Lomoriello[1] and Ivana Konvalinka[1]

[1]Section for Cognitive Systems, DTU Compute, Kongens Lyngby, Denmark
[2]Institute of Psychology, University of Regensburg, Regensburg, Germany

(iD) MZ, 0000-0001-6284-9649

We often perform actions while observed by others, yet the behavioural and neural signatures of audience effects remain understudied. Performing actions while being observed has been shown to result in more emphasized movements in musicians and dancers, as well as during communicative actions. Here, we investigate the behavioural and neural mechanisms of observed actions in relation to individual actions in isolation and interactive joint actions. Movement kinematics and EEG were recorded in 42 participants (21 pairs) during a mirror-game paradigm, while participants produced improvised movements alone, while observed by a partner, or by synchronizing movements with the partner. Participants produced largest movements when being observed, and observed actors and dyads in interaction produced slower and less variable movements in contrast with acting alone. On a neural level, we observed increased mu suppression during interaction, as well as to a lesser extent during observed actions, relative to individual actions. Moreover, we observed increased widespread functional brain connectivity during observed actions relative to both individual and interactive actions, suggesting increased intra-individual monitoring and action-perception integration as a result of audience effects. These results suggest that observed actors take observers into account in their action plans by increasing self-monitoring; on a behavioural level, observed actions are similar to emergent interactive actions, characterized by slower and more predictable movements.

# 1. Introduction

Audience effects, defined as 'a change in behaviour caused by being observed by another person' [1] are widespread and provide insight into how we socially integrate others into our action plans and outcomes. Audience effects can be distinguished from social facilitation, which are effects merely due to the presence of another person, but do not require the observed person to know—or believe—that he or she is actually being observed by this conspecific [2,3]. While a large body of research has investigated the cortical mechanisms of action observation [4–7], the behavioural and neural mechanisms underlying observed actions are heavily understudied and have not mechanistically been compared with interactive social interaction (i.e. joint action). In this study, we investigate these intra-individual mechanisms in relation to both individual actions and, importantly, interactive joint actions.

Performing actions under the watching eyes of passive observers has been previously shown to result in modulated motor behaviour. For example, dancers and musicians produce more intensified movements in front of real or virtual audiences compared with rehearsal performances [8–10]. Similarly, people put more emphasis on specific aspects of their actions when the intention of the action is to communicate information, by adjusting the timing and amplitude of actions [11], which is further influenced by the assumed age of the recipient [12]. Increased movement amplitudes have also been observed in demonstrators of xylophone players when watched by a learner [13], and in communicative actions [14]. Using communicative pointing, it was shown that motor plans of communicative, observed actions are tuned to the addressee, by increasing the informational value of movements when seen from the addressee's point of view [15]. Audience effects on actions are not restricted to demonstrators, but also to followers. Specifically, followers imitate demonstrated movements more accurately when they are watched by the demonstrator [16].

In addition to motor actions, audience effects have also been shown repeatedly in mental and moral judgement tasks. Specifically, performance (e.g. accuracy on a maths test) on easy tasks has been shown to improve, whereas performance on difficult tasks has been shown to decrease when a task is solved while the responder is observed [17]. Meanwhile, moral decisions (e.g. amount of donated money for charity) tend to be more generous and prosocial when taken under observation [18–20]. These effects have been explained by self-monitoring and self-regulatory processes related to the responder's self-image ([21,22]; for a review see [1]). The similarity of behavioural effects suggests that actors incorporate observers into their action plans when performing actions in front of an audience. However, the behavioural and neural mechanisms remain unclear.

Action observation and observed actions are thus central to social interaction and may be considered unidirectional cases of joint actions. It is well established that joint action requires close coordination of actions in space and time with those of the action partner [23]. Such coordination can be achieved by making oneself more predictable by reducing variability in one's actions [24] as well as predicting the future actions of one's partner and mutually adapting [25]. Integration of action partners is thought to be achieved by internal simulation and co-representation of the partner's actions [26–28], which has been characterized by increased mu suppression during interaction [29–31]. Mu suppression refers to the amplitude suppression of oscillations in the alpha (8–13 Hz) and beta (15–25 Hz) range over sensorimotor areas [32]. Suppression of these oscillations reflects a disinhibition of the sensorimotor system during planning, execution, perception and prediction of own and others' actions [33–37]. Therefore, if we hypothesized integration and co-representation of the observer as an action partner during observed action, we would expect to see increased mu suppression in the observed actor, as well as during bidirectional interaction.

While coupled interaction increases interpersonal dependency [38,39], recent theoretical work suggests that performance of independent actions in interaction with others requires increased self-monitoring, which may be characterized by increased internal action-perception coupling during individual actions in a social context [40]. Conversely, interpersonal coordination of actions requires mutual prediction and adaptation, resulting in the increased between-personal coupling of action-perception loops and decreased within-personal coupling (i.e. self-decoupling), presumably through a dynamic process of self–other integration [40,41]. Previous work has shown that prefrontal, premotor and parietal cortices are increasingly involved when participants attend to their own actions, as well as when they lead interactions [31,42]. Furthermore, attention to one's own actions has shown increased effective connectivity between prefrontal and premotor regions [43]. Following from this, we speculate that increased monitoring of one's own actions in a social context is supported by higher fronto-parietal and prefrontal-premotor intra-brain connectivity, while self–other integration in joint actions is supported by decreased intra-brain connectivity—the latter representing evidence of self-decoupling during self–other integration on a neural level.

Here, we investigate the intra-personal behavioural and neural dynamics underlying the performance of observed actions, in relation to individual (unobserved) actions and (coupled) interactions. We employ a mirror-game paradigm [44], which is an experimental paradigm mimicking theatre improvisation. Participants produced improvised movements alone in an *individual condition*, while observed by a partner in an *observed action condition*, or by synchronizing movements with a partner in an *interactive condition*. We speculate that movement modulations by observed actors, to some extent, resemble the surge for increased predictability and recognizability that can also be observed in social interactions. Moreover, we expect increased mu suppression in the observed actor, relative to an unobserved one, which would reflect either that the observer, despite not being overtly active, is represented in the observed actor's motor system; or increased attention to the task, in this case, attention to self-produced actions. Finally, we hypothesize a need for increased self-monitoring when being observed, characterized by increased functional connectivity between fronto-parietal and prefrontal-premotor regions, while mutual interaction results in decreased self-monitoring and consequently decreased intra-individual functional connectivity. This, we speculate, is a consequence of mutual coupling between two interacting partners, pulling us into a mutual state of interaction, away from one's own natural rhythms.

# 2. Methods

## 2.1. Participants

Fifty healthy, right-handed participants (13 female, 37 male), aged 20–33 (mean ($M$) = 24.36 years, standard deviation (s.d.) = 2.71 years), gave written informed consent to participate in the study. Participants formed 25 dyads (11 mixed-sex, 14 same-sex, incl. 1 female–female, 13 male–male). Dyads consisted of strangers ($N = 9$) or friends/partners ($N = 16$). Three dyads and two additional individual participants were excluded for technical problems during the experiment and EEG data quality issues. Therefore, 42 participants were included in the study. All participants were recruited at the Technical University of Denmark (DTU) using flyers at campus sites and online advertisement boards. The study was conducted according to the Declaration of Helsinki and was approved by DTU Compute's Institutional Review Board (COMP-IRB-2020–02). Participants were financially compensated for their participation.

## 2.2. Task

The experimental task was based on the mirror-game paradigm [44]. Two participants were seated at a table, facing each other. Each participant had a slider that could be moved laterally by approximately 20 cm on a fixed rail (figure 1a,b). Participants were asked to produce interesting and (depending on the task condition) synchronized movements, in three different conditions. Specifically, in the individual action condition, participants were asked to produce movements independently of their partner; in the interactive action condition, participants were asked to produce synchronized movements without a designated leader; and in the observed action condition, one participant was asked to produce movements as in the individual action condition while the other participant observed these movements without moving. The task included additional conditions (i.e. movement control task, leader–follower), which are not part of the current report.

A separator wall was placed between participants to block vision of their partners' face and body. A liquid crystal screen (Smart-film; CoolColour, Denmark) in the wall allowed vision of the partners' slider and hand depending on the condition requirements (i.e. closed during individual actions, otherwise open; figure 1a). The screen could be switched between transparent and opaque, to allow or prevent vision of the partners' movements. Additionally, a computer screen was inserted in the table to present instructions and additional stimuli (for the unreported movement control task) to each of the participants separately.

To reduce movement artefacts on the EEG recordings, participants were asked to restrict movements to wrist and finger movements, therefore limiting arm (and head/torso) movements. The slider was approximately 3 cm wide and had a moulded cushion for comfortable and stable finger placement. Ball bearings between the slider and rail ensured smooth and easy slider movements. The rail was mounted onto the table, with rubber patches used to dampen vibrations between components.

The experiment consisted of a total of 80 trials across five conditions (two of which are not included in the current report). Each condition was repeated 16 times; however, each participant produced movements in eight trials and observed the other in eight trials, for the observed action condition. Each trial lasted 25 s, during which participants performed the task. Between trials there were breaks

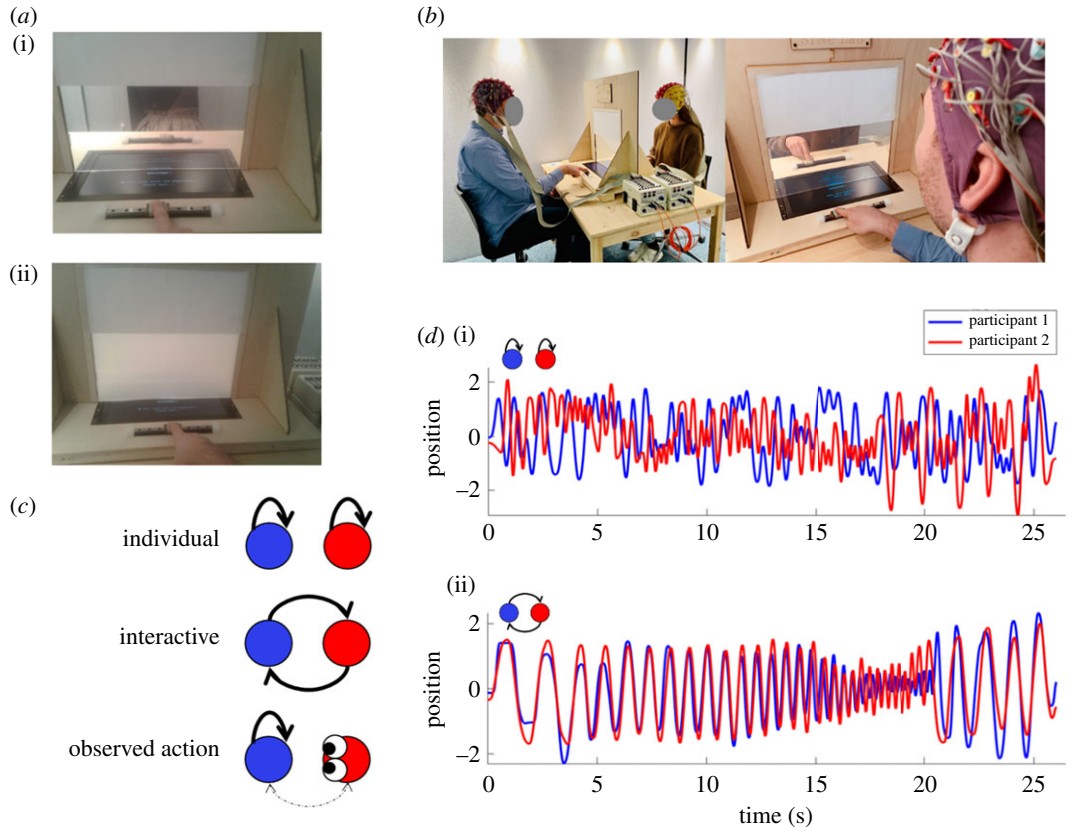

**Figure 1.** Task set-up and movement examples. (*a*) Participants' view on the task, with open (i) and closed (ii) vision of the partner. (*b*) Full EEG task set-up. (*c*) Symbolic representation of conditions with individual (no inter-subject influence), interactive (bidirectional influence) and observation condition (potential partial influence). (*d*) Example movement trajectories for individual (i) and interactive (ii) trials.

of 10 s, with longer, self-paced breaks following trials after each quarter of the experiment (see below for details, '*Subjective measures*'). Trial order was split over 16 blocks with five trials—one per condition—per block. Trial order was pseudorandomized within each block, such that the first trial in each block could not be the same condition as the last trial in the preceding block, preventing repetition of conditions in succeeding trials. At the beginning and end of the experiment, resting-state EEG was recorded during 120 s resting periods. During these periods, participants were asked to fixate on a fixation cross presented on the screen.

Each trial was preceded by a short instruction on the table screen, indicating the condition and role for each participant where necessary (i.e. *observer* or *observed actor*). A countdown indicated the start of the trial.

During the task, participants were not allowed to talk to each other or communicate in any other way except through slider movements. To prevent auditory feedback and interference from the partner's slider movements, participants wore ear plugs. Additional white noise was played in the EEG booth via room speakers.

## 2.3. Subjective measures

In every fourth block, participants were asked to rate their experienced togetherness of the preceding trial, for each trial in that block (a total of four times/condition), based on previous mirror-game experiments by Noy *et al.* [45]. Specifically, participants were prompted with the instruction '*please rate the experienced togetherness during the previous trial*'. Responses were given on a 5-point scale (1 = low, 5 = high) using a keyboard and could not be seen by the partner. Following the experimental task, participants provided information regarding their engagement with team sports, as well as dancing and practising music. Furthermore, participants were asked to categorize their relationship with their dyad partner (strangers, study acquaintances, friends/partner).

## 2.4. Data acquisition and preprocessing

EEG was recorded using two synchronized, daisy-chained 64-channel Biosemi (Amsterdam, The Netherlands) ActiveTwo EEG set-ups in a 10–20 system, at a sampling frequency of 2 kHz. Data were recorded in ActiView (v. 806) from both participants simultaneously and stored for offline analyses. EEG caps with 64 channels were positioned such that the Cz electrode was centred on the head, at the midpoint between the nasion and inion, and the left and right ear. Conductive gel (SignaGel Electrode Gel, Parker Laboratories, Farfield, NJ) was used to improve signal reception, and electrode offset was kept below 20 mV.

EEG data were processed and analysed in FieldTrip (v. 20190107, [46]) using Matlab (R2018b; The Mathworks, Natick, MA). First, EEG data were split into individual datasets. For each participant, data were demeaned, filtered (high-pass: 1 Hz; low-pass filter: 40 Hz; Butterworth filter), segmented into trials ((−3 to 25 s) relative to trial onset) and downsampled to 256 Hz. Bad channels were detected visually and removed. Muscle artefacts and other artefacts were detected semi-automatically on unfiltered data (using FieldTrip's recommendations for muscle artefact removal in ft_artifact_zvalue) and verified by visual inspection in FieldTrip. Independent component analysis (ICA; runica method) was used to detect components corresponding to eye movements and eye blinks as well as muscle artefacts. Data segments that contained artefacts were not included in the data used for ICA. Those components were regressed out of the data. Channels that were removed prior to ICA were interpolated using the average signal of neighbouring channels, and the resulting data were re-referenced to the whole-brain average. Trials where participants did not follow the instructions as indicated from the movement data were discarded from the analyses (3.3%; mostly when participants moved the slider when instructed to observe their partner's actions).

Movement data were recorded using the Polhemus LIBERTY system (Polhemus, Colchester, VT). For each participant, a movement sensor (Polhemus Micro Sensor 1.8) was placed on the tip of the right index finger (finger nail), recording the position of the finger at a sub-millimetre resolution (0.76 mm RMS). Movement data were recorded at 240 Hz and stored for offline analyses. Data synchronization was achieved via a BioSemi USB trigger interface. The distance between the source and sensors was kept below 80 cm. Movement data were processed in Matlab using in-house scripts. For each participant, missing samples were interpolated using shape-preserving piecewise cubic interpolation ('pchip' method in interp1, Matlab). Next, position over time was transformed into one-dimensional data, and data were segmented into trials and smoothed using a 100 ms centred moving average.

## 2.5. Data analyses

Alpha levels for all comparisons and tests for assumptions for ANOVA tests were set to $p < 0.05$ unless noted otherwise. It should be noted that data from both participants from all pairs (unless excluded for other reasons as mentioned above) were included in all analyses. Given that, in the observed action condition, all participants acted both as an actor and as an observer, respectively (i.e. across half of the trials, participant A observed participant B, while in the other half of the trials participant B observed participant A), each participant produced data as an observed actor in this condition, as well as an interacting partner in the interactive condition. Consequently, no selection or averaging over partners in dyads was required, and all participants were included as individual observations. No data from trials in which a participant acted as an observer were included in the analyses presented here.

Movement properties were analysed in Matlab using in-house scripts. For each trial, we calculated the movement amplitude, dominant movement frequency, movement frequency variance and jerk. Specifically, *movement amplitude* was measured using average peak-to-peak envelopes estimated using spline interpolation over local maxima separated by at least 3.5 s (Matlab function *envelope*, method: *peak*). The time window was chosen to provide smooth estimates over all conditions. Time-frequency analyses using the 'wavelet method' based on Morlet wavelets were used to obtain a time-frequency representation for each participant and each trial. Wavelet analyses were performed for each sample using wavelet widths of three cycles over frequencies between 0.25 and 20 Hz. The *dominant movement frequency* was defined as the frequency with the highest power, averaged over the time windows. *Movement frequency variance* was defined as the standard deviation of the dynamic dominant frequency within each time window. *Smoothness of movement trajectories* was calculated as jerk, i.e. the third derivative of the position signal (lower jerk = smoother trajectory); the root mean square was taken for each time window. An initial analysis that included time as an additional independent factor (based on three time windows of 7 s duration) showed no significant interactions between time

and condition for any of the measures. Therefore, time was excluded as factor for the remaining analyses. Movement parameters were compared between conditions using one-way ANOVAs and followed up with Bonferroni-corrected paired $t$-tests where applicable (i.e. in the case of significant effects at $p < 0.05$).

To test assumptions for ANOVAs, normality was tested using Kolmogorov–Smirnov tests for all movement parameters with an alpha level set to $p < 0.01$, given the moderate sample size [47]. The lowest $p$-value among the three conditions was considered to evaluate normality for each parameter. Values for the dominant movement frequency (lowest $p < 0.001$), movement frequency variability (lowest $p < 0.001$) and smoothness of movement trajectories (lowest $p = 0.002$) were not normally distributed for all conditions. Log transformation of these scores resulted in normal distribution for all conditions (all $p > 0.01$). Sphericity could not be assumed for the dominant movement frequency (Leven's statistic; $L_{(2,123)} = 4.10$, $p = 0.019$). Therefore, degrees of freedom were adjusted based on the lower-bound estimate ($\varepsilon = 0.5$). For the remaining variables, sphericity could be assumed (all $p > 0.1$).

### 2.5.1. Frequency power modulations

Frequency analyses were applied to all trials using multi-taper time-frequency transformation based on convolution in the frequency domain (mtmconvol), using Hanning windows over 1 s time windows. Frequency power was estimated for frequencies ranging from 1 to 30 Hz, at every 500 ms over the movement period, from −2 to 25 s relative to trial onset. Following frequency analysis, data segments that contained artefacts (see preprocessing) were removed (rewritten to NaN), including extended buffers of 1 s before and after each artefact onset/offset. Average time-frequency spectra were calculated for each of the conditions.

Given our particular interest in modulations of mu and occipital alpha desynchronization, we extracted frequency power from a set of predefined regions of interest reflecting sensorimotor and occipital activity. Specifically, we extracted signals from electrodes over the left and right sensorimotor cortex (C3/C4) and over the occipital cortex (average over Oz/O1/O2). Frequency power over specific frequency bands (sensorimotor: 10 Hz (8–13 Hz), 20 Hz (15–25 Hz); occipital alpha: 8–13 Hz) were calculated for the respective regions of interest in the time window of 2 to 23 s post-trial onset, averaging over time. Edges at the beginning and end of each trial were removed to allow for engagement of movements and interactions (trial onset), and to reduce noisy sections towards the end of the trial. Modulations of mu suppression and occipital alpha across the three conditions were analysed using one-way ANOVAs, and where applicable (in the case of significant $F$-test with $p < 0.05$), followed up using Bonferroni-corrected pairwise comparisons between the three conditions. Log transformation was applied to all scores as the normality of the distributions could not be assumed consistently. Following log transformation, normality could be assumed for all variables (all $p > 0.10$). Sphericity could be assumed for all comparisons (all $p > 0.10$).

### 2.5.2. Source space analysis

Source space analyses were performed to localize cortical sources of potential differences. Linearly constrained minimum variance beamformers with standard boundary element method headmodels and 10–20 electrode templates were used to estimate the time series for 70 cortical regions based on a reduced automated anatomical labelling (AAL) atlas [48], excluding subcortical and cerebellar regions (see electronic supplementary material, table S1 for full list of regions). Specifically, time series were estimated for a $1 \times 1 \times 1$ cm grid for all possible positions in the brain and summarized for each AAL region using the largest eigenvalue. Subsequently, frequency analyses were performed using the same settings as for sensor-space analyses (see above). One-way ANOVAs corrected for multiple comparisons (Bonferroni) were used to identify regions with frequency power differences between conditions.

### 2.5.3. Functional connectivity

We analysed differences in EEG functional connectivity between conditions using the weighted phase lag index (wPLI; [49]) in FieldTrip. Based on the preprocessed EEG data, we used a Laplacian spatial filter in order to reduce volume conductance effects [50] and calculated the weighted phase lag index for all channel combinations ($64 \times 64$ channels) in the alpha frequency band over non-overlapping segments of 3 s (i.e. seven evenly spaced segments per trial). Cluster permutation tests were applied to detect clusters of neighbouring connections corresponding to significant differences between conditions, using a cluster forming threshold of $p < 0.05$ and 10 000 permutations [51]. Connections were

categorized as neighbours when both ends of the connection were among the neighbouring (or the same) electrodes, adapted from the neighbourhood rules previously developed by Dumas and colleagues for inter-brain analyses [38]. For example, the connection between C3 and Oz was considered a neighbour of the connection C1-O1, since electrodes at both ends of the connection are neighbours (C3 is a neighbour of C1 and Oz is a neighbour of O1); as well as of the connection C3-O1, since O1 is a neighbour of Oz, and C3 is shared in both connections. However, Cz and Oz (or Cz-O1) is not a neighbouring connection of C3-Oz, as Cz and C3 are not neighbours or the same electrode. Consequently, if the differences between conditions passed the cluster forming threshold for multiple neighbouring connections, these were grouped together in a cluster for subsequent permutation testing.

# 3. Results

## 3.1. Movement properties

Movement synchronization was high in the interactive condition and absent in the other conditions (see electronic supplementary material), suggesting that participants succeeded in performing the task together. In the following, we compared observed actions with interactive and individual actions in terms of movement amplitude, dominant movement frequency and frequency variations, as well as smoothness of movement trajectories. Observed actions had larger amplitudes than non-observed actions. Observed actions and interactions had lower and less variable movement frequencies than individual actions. No significant differences have been observed between dyads formed by friends and dyads formed by strangers (electronic supplementary material, S5).

The movement amplitude (figure 2a) was affected by the type of social condition $F_{2,82} = 38.908$, $p < 0.001$; $\eta^2 = 0.487$; figure 2a). Specifically, movement amplitude was decreased for the interactive condition ($M = 2.57$ cm, s.d. $= 0.75$ cm) relative to the individual condition ($M = 3.04$ cm, s.d. $= 0.77$ cm; $t_{(41)} = 4.634$, $p < 0.001$) and increased for the observed action condition ($M = 3.26$ cm, s.d. $= 0.77$ cm) relative to both individual $t_{(41)} = -4.066$, $p < 0.001$ and interactive conditions $t_{(41)} = -8.942$, $p < 0.001$).

Log-transformed dominant movement frequency (figure 2b) was affected by the type of social condition $F_{2,82} = 15.497$, $p < 0.001$, $\eta^2 = 0.274$). Specifically, dominant movement frequency was decreased in the interactive condition (geometric mean (GM) $= 0.53$ Hz, geometric standard deviation factor (GSD) $= 1.47$) compared with the individual (GM $= 0.72$ Hz, GSD $= 1.42$; $t_{(41)} = 4.407$, $p < 0.001$) and observed action conditions $t_{(41)} = -2.414$, $p = 0.020$). It was also decreased for the observed action condition (GM $= 0.61$ Hz, GSD $= 1.28$) compared with the individual condition ($t_{(41)} = 4.992$, $p < 0.001$).

Log-transformed movement frequency variation (figure 2c) was affected by the type of social condition ($F_{(2.82)} = 32.751$, $p < 0.001$, $\eta^2 = 0.444$). Specifically, movement frequency variability was reduced for the interactive condition (GM $= 0.19$ Hz, GSD $= 1.76$ Hz) relative to the individual condition (GM $= 0.35$ Hz, GSD $= 1.54$ Hz; $t_{(41)} = 6.636$, $p < 0.001$) and observed action condition ($t_{(41)} = 4.755$, $p < 0.001$). It was also reduced for the observed action condition (GM $= 0.29$ Hz, GSD $= 1.50$ Hz) relative to the individual condition ($t_{(41)} = 4.881$, $p < 0.001$).

The smoothness of movement trajectories (measured as jerk; figure 2d) was affected by the type of social condition ($F_{(2,82)} = 68.869$, $p < 0.001$, $\eta^2 = 0.627$). Specifically, log of jerk was lower in the interactive condition (GM $= 0.024$ cm s$^{-3}$, GSD $= 1.43$) compared with the individual condition (GM $= 0.037$ cm s$^{-3}$, GSD $= 1.44$; $t_{(41)} = 8.683$, $p < 0.001$) and observed action condition (GM $= 0.036$ cm s$^{-3}$, GSD $= 1.43$; $t_{(41)} = 8.932$, $p < 0.001$). There was no significant difference between the individual and observed action conditions ($t_{(41)} = 1.239$, $p = 0.222$).

## 3.2. Experienced level of togetherness

Subjective ratings of 'experienced togetherness' were affected by the type of social condition ($F_{(2,75)} = 18.04$, $p < 0.001$, $\eta^2 = 0.325$), evaluated on a 5-point scale (1 = lowest, 5 = highest). Specifically, rated togetherness was higher in the interactive condition ($M = 4.05$, s.d. $= 0.70$) compared with the individual condition ($M = 3.05$, s.d. $= 1.14$; $t_{(37)} = 5.76$, $p < 0.001$) and observed action condition ($M = 3.72$, s.d. $= 0.73$; $t_{(37)} = 3.84$, $p < 0.001$). There was no significant difference between the individual and observed action conditions ($t_{(37)} = 1.52$, $p = 0.138$). Moreover, there was no significant difference between the actor and the observer within the observed action condition (actor: $M = 3.11$, s.d. $= 1.31$; observer: $M = 3.39$, s.d. $= 1.18$; $t_{(37)} = 0.71$, $p = 0.483$).

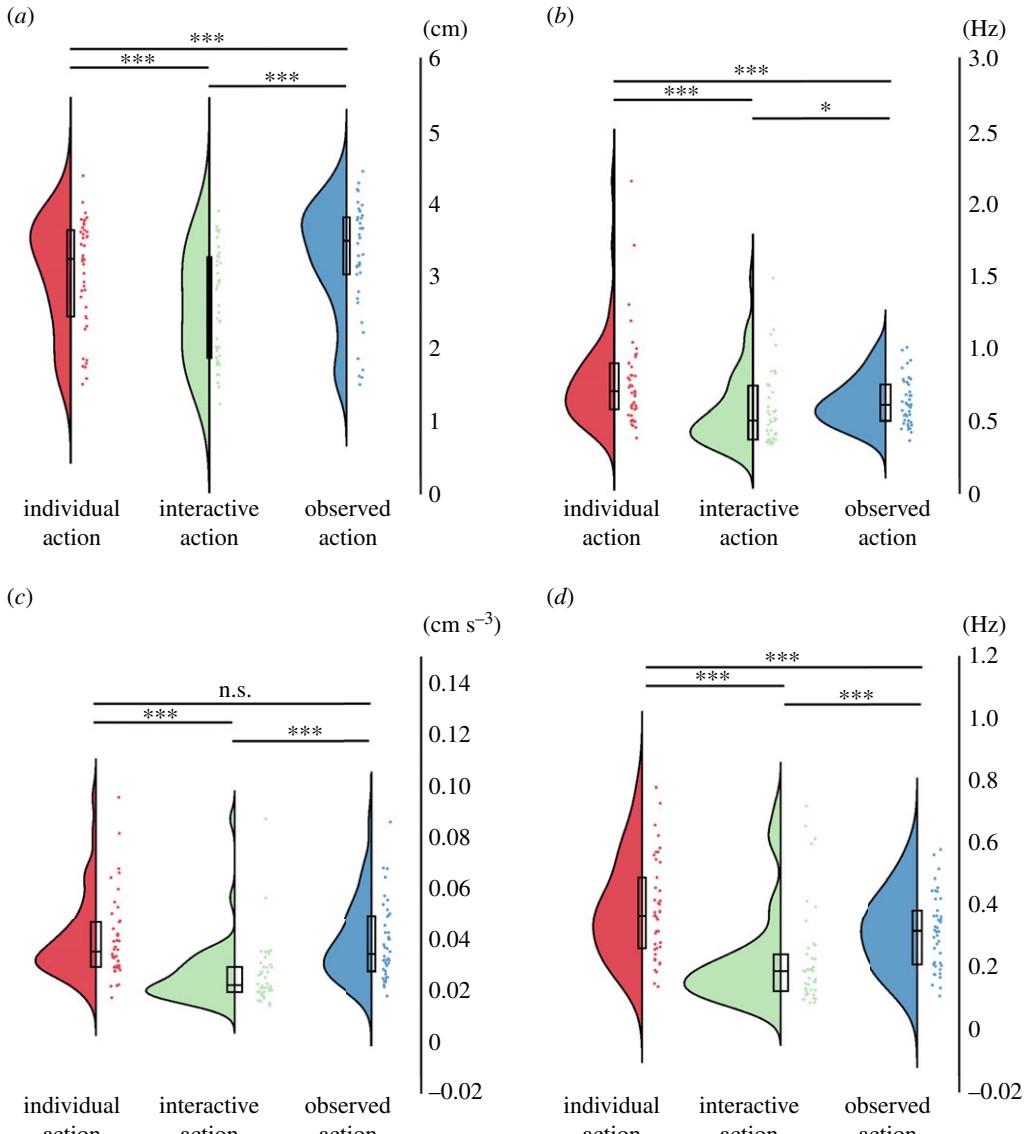

**Figure 2.** Raincloud plots of movement properties. (*a*) Movement amplitude, indicating the lateral extend of movements. (*b*) Dominant movement frequency. (*c*) Smoothness of movement trajectory ( jerk, inverted). (*d*) Variability (standard deviation) of dominant movement frequency, indicating speed changes during trials. Dots indicate participant averages. All scores are displayed as raw (non-transformed) values. Asterisks indicate level of significance (*** $p < 0.001$, ** $p < 0.01$, * $p < 0.05$); n.s.: not significant. Plots created using RainCloudPlots v. 2 [52].

## 3.3. EEG task effects

Mu suppression was observed during the individual, interactive and observed action conditions relative to pre-trial baseline (figure 3). Occipital alpha increased for individual and observed action conditions, and decreased for the interactive condition, relative to pre-trial baseline. No significant differences have been observed between dyads formed by friends and dyads formed by strangers (electronic supplementary material, S5).

## 3.4. EEG region of interest results

### 3.4.1. Sensorimotor mu (10 Hz centred component)

We analysed suppression of the 10 Hz centred sensorimotor mu component by calculating frequency power over electrodes C3 and C4 in the 8–13 Hz range (figure 4). Over the left central electrode C3, log frequency power was significantly modulated by condition $F_{(2,82)} = 29.61, p < 0.001, \eta^2 = 0.419$). Specifically, there was stronger suppression in the interactive condition compared with the individual condition ($t_{(41)} = 6.60$,

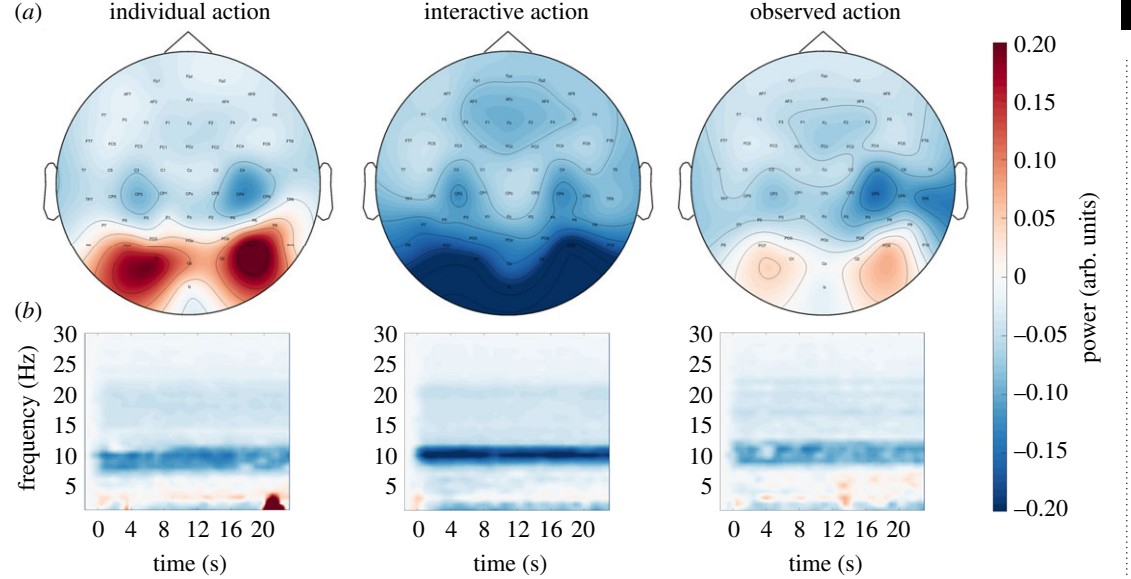

**Figure 3.** Modulation of alpha-band frequency power relative to pre-trial baseline (mu suppression). (*a*) Topoplot of alpha-band suppression during trials (8–12 Hz, absolute difference to pre-trial baseline [−2–0 s]) (*b*) TFR showing relative power to pre-trial baseline over trial time for left central electrodes (C3) per condition. Red colours indicate increase; blue colours indicate suppression, relative to pre-trial baseline. All values in arbitrary units.

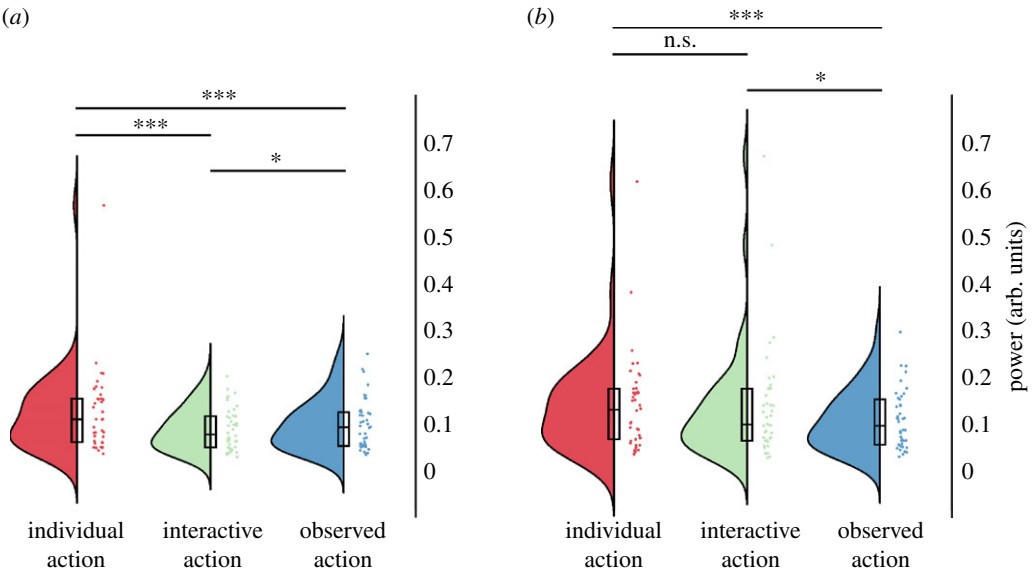

**Figure 4.** 10 Hz sensorimotor mu component power. (*a*) Left central electrodes, C3. (*b*) Right central electrodes, C4. Dots indicate participant averages. All scores are displayed as raw (non-transformed) values. Asterisks indicate level of significance (\*\*\* $p < 0.001$, \*\* $p < 0.01$, \* $p < 0.05$); n.s.: not significant. Plots created using RainCloudPlots v. 2 [52].

$p < 0.001$) and observed action condition ($t_{(41)} = -2.96$, $p = 0.015$). There was also a stronger suppression during the observed action condition compared with the individual condition ($t_{(41)} = 5.10$, $p < 0.001$). Frequency power was also modulated over the right central electrode, C4 ($F_{(2,82)} = 14.14$, $p < 0.001$, $\eta^2 = 0.256$). It did not differ significantly between individual and interactive conditions ($t_{(41)} = 2.01$, $p = 0.155$), but was suppressed in the observed action condition when compared both with the individual ($t_{(41)} = 6.66$, $p < 0.001$) and interactive conditions ($t_{(41)} = 3.05$, $p = 0.012$; all Bonferroni corrected).

### 3.4.2. Sensorimotor mu (20 Hz centred component)

Log frequency power in the 20 Hz centred sensorimotor mu component was affected by condition for the left hemisphere (C3; $F_{(2,82)} = 6.87$, $p = 0.002$, $\eta^2 = 0.144$; figure 5*a*), as well as the right hemisphere (C4;

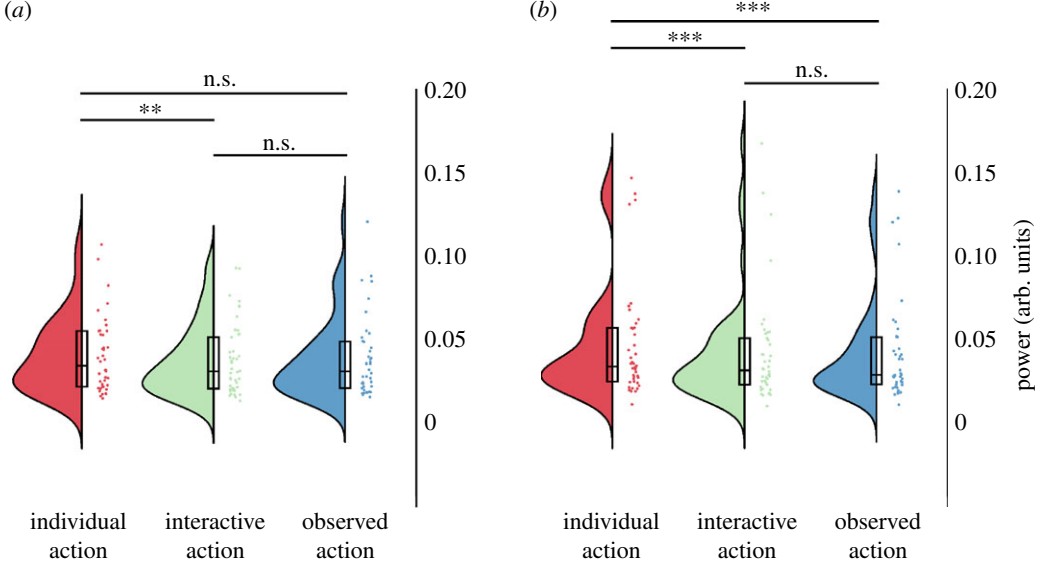

**Figure 5.** 20 Hz sensorimotor mu component power. (a) Left central electrodes, C3. (b) Right central electrodes, C4. Dots indicate participant averages. All scores are displayed as raw (non-transformed) values. Asterisks indicate level of significance (*** $p < 0.001$, ** $p < 0.01$, * $p < 0.05$); n.s.: not significant. Plots created using RainCloudPlots v. 2 [52].

$F_{(2,82)} = 6.65$, $p = 0.002$, $\eta^2 = 0.140$; figure 5b). Over the left central electrode, C3, frequency power was more suppressed in the interactive condition compared with the individual condition ($t_{(41)} = 3.99$, $p = 0.001$). Frequency power in the observed action condition was in between individual and interactive conditions, and did not differ significantly from either (individual: $t_{(41)} = 2.30$, $p = 0.079$; interactive: $t_{(41)} = -0.80$, $p = 1$; all Bonferroni corrected). Over right central electrode, C4, frequency power was more suppressed during the interactive condition compared with individual condition ($t_{(41)} = 5.16$, $p < 0.001$) as well as observed action condition compared with individual condition ($t_{(41)} = 4.73$, $p < 0.001$). There was no significant difference between the observed action condition and the interactive condition ($t_{(41)} = 0.04$, $p = 1$; all Bonferroni corrected).

### 3.4.3. Occipital alpha

Suppression of log frequency power of occipital alpha (8–12 Hz, averaged over Oz, O1 and O2; figure 6) was significantly affected by condition ($F_{(2,82)} = 34.68$, $p < 0.001$, $\eta^2 = 0.458$). Specifically, the suppression of occipital alpha was stronger during the interactive compared with individual action condition ($t_{(41)} = 6.55$, $p < 0.001$). Occipital alpha suppression in the observed action condition was at an intermediate level between the individual and interactive conditions. Specifically, the suppression was stronger relative to the individual condition ($t_{(41)} = 5.64$, $p < 0.001$) and weaker compared with interactive condition ($t_{(41)} = -4.39$, $p < 0.001$; all Bonferroni corrected).

*Whole-brain source space frequency analyses* were conducted for cortical regions (based on the AAL template, see Methods), in order to identify potential differences in other cortical regions. After correction for multiple comparisons (Bonferroni, 70 regions), we observed significant effects of condition on alpha-band power in frontal (precentral R, superior frontal R, supplementary motor area (SMA) R), occipital (calcarine sulcus L/R, cuneus L/R, lingual gyrus L/R, occipital cortex sup/mid/inf L/R, fusiform gyrus L/R), parietal (superior parietal L, angular gyrus L/R, precuneus L/R) and temporal (temporal mid/inf L/R) regions (one-way ANOVA, all $p < 0.05$, Bonferroni corrected). Specific differences between conditions are listed in the electronic supplementary material, S4. The results suggest occipital sources (strongest suppression in interactive actions, followed by observed actions), as well as temporal sources in the interactive and observed action condition (both stronger suppression compared with the individual condition, but no differences between the interactive and observed action conditions), as well as frontal suppression during individual and observed action conditions compared with interactive condition.

In the beta band, only the right SMA showed reliable significant differences after correction for multiple comparison ($F_{(2,82)} = 9.32$, $p = 0.016$, $\eta^2 = 0.185$). Beta band power in SMA was higher in the individual compared with the interactive action condition ($F_{(1,41)} = 14.56$, $p < 0.001$, $\eta^2 = 0.262$) and

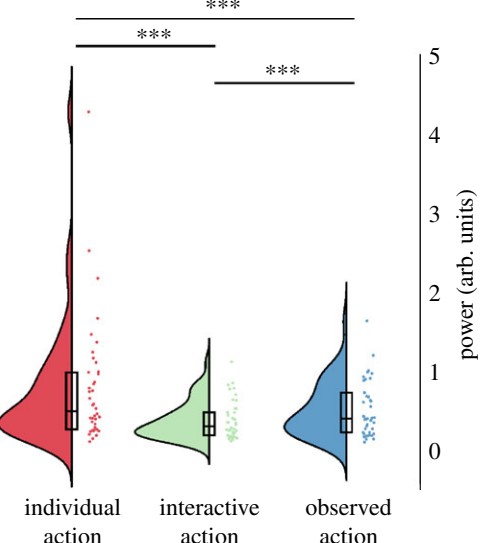

**Figure 6.** Occipital alpha. Dots indicate participant averages. All scores are displayed as raw (non-transformed) values. Asterisks indicate level of significance (*** $p < 0.001$, ** $p < 0.01$, * $p < 0.05$); n.s.: not significant. Plots created using RainCloudPlots v. 2 [52].

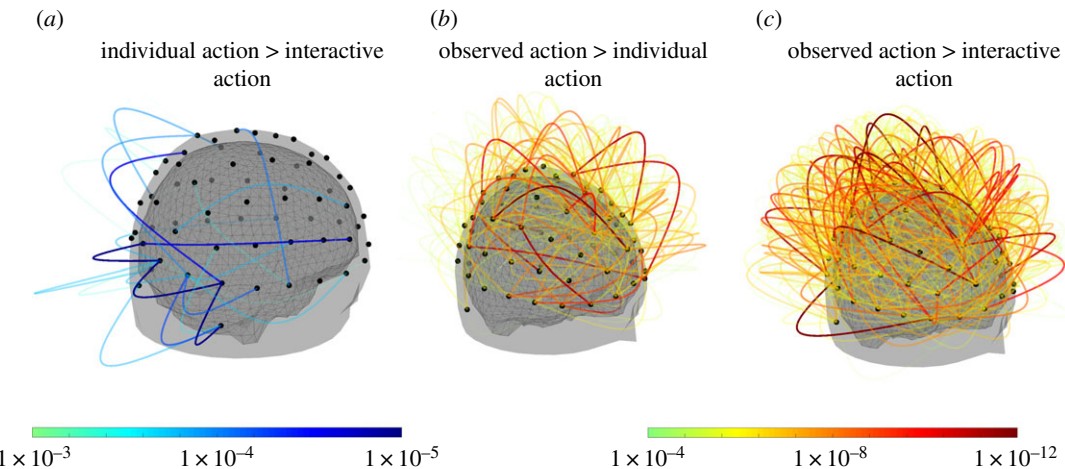

**Figure 7.** Functional connectivity in the alpha-band. Clusters of functional connectivity (wPLI) in the alpha frequency band corresponding to differences between conditions. (*a*) Stronger functional connectivity in the individual action condition compared with the interactive action condition. (*b*) Stronger functional connectivity in the observed action condition compared with the individual action condition. (*c*) Stronger functional connectivity in the observed action condition compared with the interactive condition. Black dots indicate electrode positions (64 channels); line colours indicate link strength (*p*-value).

observed action condition ($F_{(1,41)} = 6.52$, $p = 0.015$, $\eta^2 = 0.137$). There were no significant differences between the interactive and observed action conditions ($F_{(1,41)} = 3.49$, $p = 0.069$, $\eta^2 = 0.078$).

## 3.5. EEG functional connectivity

A pairwise cluster permutation analysis of weighted phase lag index differences revealed significant effects of the type of social condition on functional connectivity in the alpha-band. Specifically, the individual action condition showed significant differences from the interactive condition ($p < 0.001$), with a corresponding cluster including occipital, parietal and temporal connections (figure 7a). The observed action condition showed significant differences from both individual and interactive conditions (both $p < 0.001$), with corresponding widespread clusters spanning most electrodes (individual versus observed actions: 58.7% of all possible links between electrode pairs; interactive versus observed actions: 81.4% of all possible links). The connectivity cluster of the observed action condition relative to the individual condition was most pronounced between frontal and

centro-parietal electrodes (figure 7b); functional connectivity in the observed action condition relative to the interactive condition was most pronounced between frontal and occipital electrodes (figure 7c).

# 4. Discussion

In this study, we aimed to investigate the behavioural and neural mechanisms underlying observed actions, corresponding to audience effects [1]. We observed that participants, when observed by a partner, produced larger movements than during non-observed actions, as well as during interactions where two partners strived to synchronize their movements. Both observed actors and interacting partners produced slower and less variable movements in contrast with acting alone, with interacting partners achieving high levels of synchronization. Using EEG, we observed stronger mu suppression during interaction compared with individual actions, which was also present, albeit weaker, during observed actions. Moreover, we observed an increase in widespread functional brain connectivity in the alpha frequency range during observed actions.

Our results show that actions are performed differently when they are observed by another person, even in the absence of a communicative task, following the framework of audience effects [1]. Specifically, our results suggest that observed actions resemble emergent interactive actions in movement properties, and that observed actors take observers into account in their action plans.

## 4.1. Changes in movement properties depending on the type of social condition

Observed actors produced larger and more regular movements than when acting alone. These modulations fit largely with the idea that movements are intensified when performed under the watching eyes of an observer [8–10]. Additionally, by reducing the variation in movement frequency throughout a trial, as well as by reducing movement frequency overall, observed actors produced actions that were more predictable to the observer, and thus easier to follow, despite the fact that observers did not have the task of responding to or predicting the observed movements. Increased predictability was also observed in coupled interactions (again by reduced frequency variation and frequency); however, here the movements were smaller in terms of amplitude. Smaller movement amplitudes allow for lower movement velocities at a constant frequency, which would allow both partners to respond to changes in movements of their partners more easily. However, during coupled interaction, the produced movements emerged from the interaction and were the consequence of mutual adaptation [25], whereas observed actions were completely self-regulated. These differences may allow observed actors to produce large movements (with necessarily higher peak velocities and accelerations), resembling communicative actions [11], whereas mutual adaptation probably benefits from slower velocities and lower acceleration rates. Modulations aimed at increased predictability, however, are probably due to incorporation of the observer, in the same way as actions are made more predictable during interactions [24].

## 4.2. Increased motor system engagement during interactive and observed actions

Mu suppression, especially over the left hemisphere (contralateral to the used effectors) in the 10 and 20 Hz component, was increased both during interaction and observed actions relative to individual actions. Increased mu suppression has been found across a range of interactions including joint action, joint attention and coordination [29–31,53,54], and has been associated with the integration of social partners in neural 'we-representations' [28] during social interaction. More broadly, it has previously been suggested that alpha oscillations have a functional role related to attention and prediction [37], with the mu-rhythm reflecting a common coding mechanism of perception and action [36,55]. With regard to the present study, increased mu suppression during the interactive condition may thus reflect increased attention to self–other coupling of action-perception loops [31], reflecting the integration of one's own movements with those of the partner in order to achieve and sustain synchronized movements. However, the role of increased mu suppression during the observed action condition is less clear. One possibility is that observed actors focus more on their own movements, resulting in more strictly planned and controlled movements, with mu suppression reflecting increased action-perception coupling of intrinsic (self-related) loops. Alternatively, the observer may be integrated into the action plans of the observed actor directly in 'we-representations' [28,56]. However, in contrast with the interactive condition, the observer is not performing any overt actions him-/

herself; therefore, it remains unclear what exactly would be integrated. One possibility may be that the observed actor (implicitly) imagines the observer to 'go along' with his or her actions, that is, to mentally synchronize their actions to the observed actor through mutual self-adaptation between the own and the simulated movements. Another possibility is that the observation process itself is integrated. It has been shown that, when observing other people's actions, the observer internally simulates the observed actions in the observer's motor system [5,57], which has repeatedly been associated with mu suppression in the observer [35,58,59]. Therefore, the observed actor may not only represent his or her own actions, but also the observation process of the observer, resulting in increased mu suppression relative to non-observed actions.

## 4.3. Increased intra-individual coupling during observed actions

We observed a decrease in functional connectivity during interactions relative to individual actions (in isolation) with corresponding clusters of connections between occipital, parietal and temporal areas in the alpha frequency range, as well as increased widespread functional connectivity during observed actions. Notably, the strongest increases in functional connectivity (during observed actions) were found between frontal and centro-parietal regions (figure 7*b*). Previous studies have shown connectivity changes between similar regions when participants attended to their own actions [43], and functional connectivity in the alpha-band has previously been tied to functions of cognitive control [60]. The prefrontal cortex is also known to be involved in higher level cognitive control of actions [61,62], suggesting an increased level of self-monitoring and, possibly, higher attention towards own actions in the observed actor.

These results thus suggest that individual actions that do not rely on integration of actions of the other result in stronger intra-individual coupling, presumably due to an increase in monitoring of own actions. Interactions, on the other hand, require a tight coupling of self and other, hence a stronger interpersonal coupling, which we speculate may result in weaker intra-individual coupling due to reduced monitoring (and perhaps regulation) of own actions. This mechanism has recently been suggested following results from a Kuramoto model of self–other integration applied to interpersonal finger tapping data [40]. Heggli and colleagues showed that interpersonal coordination strategies rely on the dynamic balance between within- and between-person coupling of action-perception units. Namely, mutual adaptation (interaction) exhibited higher interpersonal coupling and lower intra-personal coupling of action-perception units. This was in contrast with 'leader-leader' behaviour, where two people interact but do not adapt to one another—which was theoretically characterized by lower interpersonal coupling, and higher intra-personal coupling. While this is in line with the functional connectivity results in the present study, it should be noted that the functional connectivity differences here are generally widespread, instead of localized to fronto-parietal and prefrontal-premotor connections as hypothesized, which suggests intra-personal coupling effects that extend beyond action-perception loops. We thus speculate that the widespread connectivity differences we see between the conditions may also be related to other task-related differences, e.g. intensification of movement amplitudes.

## 4.4. Interpretational issues

Given the task dependency of audience effects, the results of the present study cannot be interpreted independently of the task context. Specifically, observed actions in the current study were intermixed with, among others, trials of coupled interaction and leader–follower interactions. Therefore, the adaptations of the observed actor when there is no communicative intention may be affected by the other conditions in which there was a communicative intention, or mutual adaptation. However, despite this lack of separation, similar behavioural changes could be expected in the individual condition if the observed effects during the observed action condition were merely carry-over effects from the other conditions. Therefore, we are confident that the observed effects are proper audience effects.

Furthermore, it cannot be excluded that the finding of increased mu suppression during the interaction and observed action conditions are related to attentional processes. As described by Hamilton & Lind [1], the presence of an observer—which may have occurred in both conditions equally—can affect the effort at which a task is executed. In turn, one interpretation could be that participants were less motivated in the individual condition, compared with both the interactive and observed action condition. This may have resulted in more variable, and thus less predictable movements, and consequently, lower level of sensorimotor processing. Similarly, the observed mu suppression effects could also be related to attention or cognitive demand [37], mechanisms related to

reduction in power of alpha oscillations over occipital areas [63]. Such attention/demand-related suppression of alpha oscillations may therefore be related to the increased demands on coordination of one's own movements with those of the partner in the interactive condition, as well as a perceived need to produce 'better' movements (e.g. more interesting and creative, or less sluggish) in the observed action condition. As such, these conditions would not necessarily require an integration of the partner or observer, but may simply be more challenging, either explicitly (due to being instructed to synchronize) or implicitly (due to a perceived pressure to produce 'better' movements).

Finally, most participants in the present study were improvisation novices. Improvisation novices have been shown to produce less creative movements and lower peak velocities during interactions in the mirror-game paradigm than improvisation experts, and generally perform better when one person is declared the designated leader [44]. This may have limited performance and restricted movements during the interactive condition, and possibly also during the observed action condition, resulting in more predictive and less expressive movements compared with the individual and unobserved condition.

## 4.5. Observers are integrated in observed actions

The suggestion that observers are integrated into action plans is closely related to audience effects in other situations and actions. A number of studies have shown that we adapt our behaviour and responses when we are being observed by others [8–10,13–15,17–20], or during communicative actions [11,12]. Specifically, actors adapt their communication style to the predicted needs of their partners. Hamilton & Lind [1] suggest that audience effects, in the context of mental tasks and moral/prosocial judgements and questions, are related to one's self-concept, and intended towards keeping a positive image and appearance in the observer, requiring that observed actors take into account the perspective of the observer. In contrast with previous studies, however, here we provide evidence for an integration of passive observers of manual actions, in the absence of communicative intentions. These findings have in common that actions are adjusted in response to being observed, based on an implicit simulation or consideration of the observer.

## 5. Conclusion

We have shown that when actions are performed under the watching eyes of an observer, more predictable and more intense movements are produced, that would be easier to follow by an observer. Moreover, our results suggest that these behavioural adaptations may be due to an integration of the observer into one's own action plans, as well as a stronger intra-individual neural coupling, potentially as a result of increased self-monitoring and attention towards one's own actions. These findings advance our understanding of the mechanisms underlying audience effects in manual actions and provide a complementary approach to the increasing number of research studies on inter-individual mechanisms.

Data accessibility. The (preprocessed) data analysed in this manuscript and the code used for data analysis are available in the following OSF repository: https://osf.io/5nt8f. Full raw data are available via the DTU data repository (linked in OSF).
Authors' contributions. M.Z.: conceptualization, data curation, formal analysis, investigation, methodology, project administration, software, supervision, visualization, writing—original draft and writing—review and editing; A.S.L.: data curation, investigation, writing—original draft and writing—review and editing; I.K.: conceptualization, formal analysis, funding acquisition, investigation, project administration, resources, supervision, writing—original draft and writing—review and editing. All authors gave final approval for publication and agreed to be held accountable for the work performed therein.
Competing interests. We declare we have no competing interests.
Funding. This work is supported by the Villum Experiment grant (project no. 00023213).
Acknowledgements. We would like to thank DTU Skylab for providing technical support and the workspace needed during the construction of the experimental set-up, Hearing Systems (DTU Health Tech) for use of the EEG laboratory facilities, as well as Andrew King (DTU Health Tech) for technical support in the EEG laboratory.

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
