## [Peer Review File · Royal Society Open Science]

Review History

RSOS-211352.R0 (Original submission)

Review form: Reviewer 1

Is the manuscript scientifically sound in its present form?

Yes

Are the interpretations and conclusions justified by the results?

Yes

Is the language acceptable?

Yes

Do you have any ethical concerns with this paper?

No

Have you any concerns about statistical analyses in this paper?

Yes

Recommendation?

Major revision is needed (please make suggestions in comments)

Comments to the Author(s)

The study focuses on the audience effect (i.e., the tendency of a person to adapt their behavior when being observed), and specifically tries to disentangle effects of a passive observer from an interactive partner on movements produced within the so-called mirror game. EEG is used to test whether there is more or less integration (measured as mu suppression) during more individually-focused movements compared to those produced with a partner. The authors find several kinematic features, such as movement size and variability, that are affected by being observed or being produced within an interaction, and further show that there is more intra-brain coupling (taken as more integration) during individual actions compared to observed and interactive actions.

Overall I thought this was an interesting study and a relevant addition to the field in terms of using 3 conditions to distinguish pure audience effects from joint-action/interaction effects. My questions for the authors relate primarily to clarifying some aspects of the methods, and providing a more rounded discussion of the results.

Major comments:

Is subjective measure of “Experienced togetherness” based on any previous work? If not, can the authors provide more information about what exactly the participants were asked?

The methods section describes that “Sphericity could be assumed for all comparisons (all $p < .10$).” Does this mean that the threshold for the sphericity test was set at 0.10, and no values were below this threshold? Or is this just saying that all sphericity test values were greater than 0.10? If the threshold was indeed 0.10, can the authors give some reference for this value, since it’s quite different from the typical 0.05 (or lower) threshold? Otherwise, I would recommend clarifying these statements so that the thresholds and the actual values are clearly given.

Functional Connectivity: I’m not very familiar with this type of FC analysis, but I found the description related to clustering to be a bit unclear. I think providing an example here would be helpful for the reader to know what a cluster might be.

For the EEG Task Effects results, from which participant are these results? The actor, observer, or average across both? It would be helpful to specify this, both in the methods and results sections. It seems to me that the results all come from one person in the dyad (i.e., the one performing the action, and presumably just one of those in the interaction condition). If the data indeed come from just one participant in each dyad, was anything done with the other participant's (i.e., the observer's) data?

The mu suppression results are discussed as an index of integration, but there are many studies suggesting that alpha/mu suppression indexes attention, cognitive processing demand, or something similar. Could this also be an explanation for the findings? For example, the interactive task is likely much more challenging than the other conditions given the extra need to coordinate. The observed action condition may provide more pressure to perform “interesting” movements. In both cases, one could argue that there is no need to represent the other person in the dyad, only an increased task difficulty due to “better” (more interesting) performance, or producing movement that is easier to track (when leading) or otherwise to synchronize one’s own movement with what is being observed (when following).

When discussing the intra-personal coupling results, the authors refer to Heggli and colleagues' model of self-other integration as an explanation for the increased intrapersonal connectivity observed in the present study. Can the authors elaborate a bit on how intra-personal neural coupling relates to the perception-action coupling in the Heggli model? Particularly since the EEG results are quite general (i.e., not about a specific region-to-region connection).

More generally, given that you have the dual-EEG data, I was wondering why there were no analyses of inter-brain coupling, particularly since inter-brain results are brought up in the discussion, it seems this would be quite relevant.

Minor:

In the introduction, the authors provide a good background and motivation for the study, but I noticed that the studies on audience effects and communicative intentions in action kinematics were from 2011 and before, whereas there are a few more recent papers that I think would make a good addition to the literature review here.

For instance, Krishnan-Barman & Hamilton, 2019

(<https://doi.org/10.1016/j.cognition.2019.03.007>), McEllin et al., 2018

(<http://dx.doi.org/10.1037/xhp0000505>), Trujillo et al., 2018

(<https://doi.org/10.1016/j.cognition.2018.04.003>), Winner et al., 2019 (DOI: 10.1111/cogs.12733)

Review form: Reviewer 2

Is the manuscript scientifically sound in its present form?

Yes

Are the interpretations and conclusions justified by the results?

Yes

Is the language acceptable?

Yes

Do you have any ethical concerns with this paper?

No

Have you any concerns about statistical analyses in this paper?

No

Recommendation?

Accept with minor revision (please list in comments)

Comments to the Author(s)

In general, it is a well-written manuscript, with a detailed and easy-to-follow description of all methods and results. Additionally, results are clearly presented in text and figures. I would like to additionally praise authors for sharing data. Results are finely discussed and limitations of interpretation are clearly stated. I enjoyed reading this manuscript. There are three minor suggestions/questions I have for authors. As these questions are not crucial, I suggest the manuscript for publication.

#1 (minor)

The authors talk about the audience effect to introduce their research question. I'm wondering how different is the concept of audience effect from the social facilitation effect? As far I understand these two are really similar, and social facilitation has been studied in many different behavioral tasks, where the mere presence of the co-actor influences the behavior. What are the differences between audience effect and social facilitation? I believe that answering these questions would improve the theoretical framing of the presented experiment.

#2 (minor)

The manuscript focuses on action monitoring and its neural underpinnings (measured with EEG), therefore, I would like to suggest that a classic paper showing the effect of observation during action monitoring should be referenced:

van Schie, H., Mars, R., Coles, M. et al. Modulation of activity in medial frontal and motor cortices during error observation. *Nat Neurosci* 7, 549-554 (2004).
<https://doi.org/10.1038/nn1239>

#3 (minor)

The authors report that they used two different groups of dyads (strangers and familiar (friends and couples)). There is much evidence that joint actions/social interactions are heavily affected by the relations between members of the dyad. I'm wondering why authors decided to mix different types of dyads? Maybe it should be additionally analyzed whether effects reported in the manuscript are the same for both groups (strangers vs. familiar).

Decision letter (RSOS-211352.R0)

Dear Dr Zimmermann

The Editors assigned to your paper RSOS-211352 "Intra-individual behavioural and neural signatures of audience effects and interactions in a mirror-game paradigm" have now received comments from reviewers and would like you to revise the paper in accordance with the reviewer comments and any comments from the Editors. Please note this decision does not guarantee eventual acceptance.

Please submit your revised manuscript and required files (see below) no later than 21 days from today's (ie 25-Nov-2021) date. Note: the ScholarOne system will 'lock' if submission of the revision is attempted 21 or more days after the deadline. If you do not think you will be able to meet this deadline please contact the editorial office immediately.

on behalf of Dr Jennifer Cook (Associate Editor) and Essi Viding (Subject Editor)
openscience@royalsociety.org

Associate Editor Comments to Author (Dr Jennifer Cook):

Associate Editor: 1

Comments to the Author:

Thank you for submitting your manuscript to RSOS. I have sent it out to expert reviewers who are both positive about your paper and have made some helpful comments, as you can see from the reviews. I hope you can use all the comments for a revision and then resubmit the paper. Please be sure to pay particular attention to Reviewer 1's request for further clarity regarding measures, statistics and analyses, and their recommendations for further discussion points (e.g. alternative interpretation of the mu suppression results).

Reviewer comments to Author:

Reviewer: 1

Comments to the Author(s)

The study focuses on the audience effect (i.e., the tendency of a person to adapt their behavior when being observed), and specifically tries to disentangle effects of a passive observer from an interactive partner on movements produced within the so-called mirror game. EEG is used to test whether there is more or less integration (measured as mu suppression) during more individually-focused movements compared to those produced with a partner. The authors find several kinematic features, such as movement size and variability, that are affected by being observed or being produced within an interaction, and further show that there is more intra-brain coupling (taken as more integration) during individual actions compared to observed and interactive actions.

Overall I thought this was an interesting study and a relevant addition to the field in terms of using 3 conditions to distinguish pure audience effects from joint-action/interaction effects. My questions for the authors relate primarily to clarifying some aspects of the methods, and providing a more rounded discussion of the results.

Major comments:

Is subjective measure of “Experienced togetherness” based on any previous work? If not, can the authors provide more information about what exactly the participants were asked?

The methods section describes that “Sphericity could be assumed for all comparisons (all $p > .10$).” Does this mean that the threshold for the sphericity test was set at 0.10, and no values were below this threshold? Or is this just saying that all sphericity test values were greater than 0.10? If the threshold was indeed 0.10, can the authors give some reference for this value, since it’s quite different from the typical 0.05 (or lower) threshold? Otherwise, I would recommend clarifying these statements so that the thresholds and the actual values are clearly given.

Functional Connectivity: I’m not very familiar with this type of FC analysis, but I found the description related to clustering to be a bit unclear. I think providing an example here would be helpful for the reader to know what a cluster might be.

For the EEG Task Effects results, from which participant are these results? The actor, observer, or average across both? It would be helpful to specify this, both in the methods and results sections. It seems to me that the results all come from one person in the dyad (i.e., the one performing the action, and presumably just one of those in the interaction condition). If the data indeed come from just one participant in each dyad, was anything done with the other participant's (i.e., the observer's) data?

The mu suppression results are discussed as an index of integration, but there are many studies suggesting that alpha/mu suppression indexes attention, cognitive processing demand, or something similar. Could this also be an explanation for the findings? For example, the interactive task is likely much more challenging than the other conditions given the extra need to coordinate. The observed action condition may provide more pressure to perform “interesting” movements. In both cases, one could argue that there is no need to represent the other person in the dyad, only an increased task difficulty due to “better” (more interesting) performance, or producing movement that is easier to track (when leading) or otherwise to synchronize one’s own movement with what is being observed (when following).

When discussing the intra-personal coupling results, the authors refer to Heggli and colleagues’ model of self-other integration as an explanation for the increased intrapersonal connectivity observed in the present study. Can the authors elaborate a bit on how intra-personal neural coupling relates to the perception-action coupling in the Heggli model? Particularly since the EEG results are quite general (i.e., not about a specific region-to-region connection).

More generally, given that you have the dual-EEG data, I was wondering why there were no analyses of inter-brain coupling, particularly since inter-brain results are brought up in the discussion, it seems this would be quite relevant.

Minor:

In the introduction, the authors provide a good background and motivation for the study, but I noticed that the studies on audience effects and communicative intentions in action kinematics were from 2011 and before, whereas there are a few more recent papers that I think would make a good addition to the literature review here.

For instance, Krishnan-Barman & Hamilton, 2019

(<https://doi.org/10.1016/j.cognition.2019.03.007>), McEllin et al., 2018

(<http://dx.doi.org/10.1037/xhp0000505>), Trujillo et al., 2018

(<https://doi.org/10.1016/j.cognition.2018.04.003>), Winner et al., 2019 (DOI: 10.1111/cogs.12733)

Reviewer: 2

Comments to the Author(s)

In general, it is a well-written manuscript, with a detailed and easy-to-follow description of all methods and results. Additionally, results are clearly presented in text and figures. I would like to additionally praise authors for sharing data. Results are finely discussed and limitations of interpretation are clearly stated. I enjoyed reading this manuscript. There are three minor suggestions/questions I have for authors. As these questions are not crucial, I suggest the manuscript for publication.

#1 (minor)

The authors talk about the audience effect to introduce their research question. I'm wondering how different is the concept of audience effect from the social facilitation effect? As far I understand these two are really similar, and social facilitation has been studied in many different behavioral tasks, where the mere presence of the co-actor influences the behavior. What are the differences between audience effect and social facilitation? I believe that answering these questions would improve the theoretical framing of the presented experiment.

#2 (minor)

The manuscript focuses on action monitoring and its neural underpinnings (measured with EEG), therefore, I would like to suggest that a classic paper showing the effect of observation during action monitoring should be referenced:

van Schie, H., Mars, R., Coles, M. et al. Modulation of activity in medial frontal and motor cortices during error observation. *Nat Neurosci* 7, 549–554 (2004).
<https://doi.org/10.1038/nn1239>

#3 (minor)

The authors report that they used two different groups of dyads (strangers and familiar (friends and couples)). There is much evidence that joint actions/social interactions are heavily affected by the relations between members of the dyad. I'm wondering why authors decided to mix different types of dyads? Maybe it should be additionally analyzed whether effects reported in the manuscript are the same for both groups (strangers vs. familiar).

===PREPARING YOUR MANUSCRIPT===

If you have been asked to revise the written English in your submission as a condition of publication, you must do so, and you are expected to provide evidence that you have received language editing support. The journal would prefer that you use a professional language editing service and provide a certificate of editing, but a signed letter from a colleague who is a fluent speaker of English is acceptable. Note the journal has arranged a number of discounts for authors using professional language editing services (<https://royalsociety.org/journals/authors/benefits/language-editing/>).

===PREPARING YOUR REVISION IN SCHOLARONE===

- Ensure that your data access statement meets the requirements at <https://royalsociety.org/journals/authors/author-guidelines/#data>. You should ensure that you cite the dataset in your reference list. If you have deposited data etc in the Dryad repository, please include both the 'For publication' link and 'For review' link at this stage.
- If you are requesting an article processing charge waiver, you must select the relevant waiver option (if requesting a discretionary waiver, the form should have been uploaded at Step 3 'File upload' above).
- If you have uploaded ESM files, please ensure you follow the guidance at <https://royalsociety.org/journals/authors/author-guidelines/#supplementary-material> to include a suitable title and informative caption. An example of appropriate titling and captioning may be found at https://figshare.com/articles/Table_S2_from_Is_there_a_trade-off_between_peak_performance_and_performance_breadth_across_temperatures_for_aerobic_sc_ope_in_teleost_fishes_/3843624.

Author's Response to Decision Letter for (RSOS-211352.R0)

See Appendix A.

Decision letter (RSOS-211352.R1)

Dear Dr Zimmermann,

It is a pleasure to accept your manuscript entitled "Intra-individual behavioural and neural signatures of audience effects and interactions in a mirror-game paradigm" in its current form for publication in Royal Society Open Science. The comments of the reviewer(s) who reviewed your manuscript are included at the foot of this letter.

on behalf of Dr Jennifer Cook (Associate Editor) and Essi Viding (Subject Editor)
openscience@royalsociety.org

Appendix A

Response to reviewers – Royal Society Open Science – RSOS-211352

Zimmermann M, Schiano Lomoriello A, Konvalinka I

Intra-individual behavioural and neural signatures of audience effects and interactions in a mirror-game paradigm

Dear Editors and reviewers,

We would like to thank the editors and reviewers for their helpful comments to further improve our manuscript. We agree with most comments raised by the reviewers as you will see below, where we respond to each of them on a point by point basis. While we agree that the interbrain coupling analyses are of importance, we deliberately decided to not include them in this manuscript as we think the theoretical background and methodological approach diverges from the one presented in the current manuscript. Thus, as explained below, we are writing up the interbrain/interpersonal results for a different manuscript.

In extension to the changes based on the comments by the reviewers, we made some adjustments in the description of the functional connectivity analysis and its results. Specifically, we removed the mention of connectivity analysis in the beta range (for which no results had been reported, and no hypotheses had been made) from the methods, and specified the frequency band (i.e. alpha) in the results. Moreover, we specified the connections to “*fronto-parietal and prefrontal-premotor connections*”, rather than a three-way “*prefrontal-premotor-parietal*” connectivity. The findings are discussed in line with the reviewers’ comments on the existing connectivity analysis (see below).

We are confident that the reviews have improved the manuscript and look forward to hearing back.

With kind regards,

– on behalf of all authors –

Marius Zimmermann

Reviewer comments to Author:

Reviewer: 1

Major comments:

(1) Is subjective measure of “Experienced togetherness” based on any previous work? If not, can the authors provide more information about what exactly the participants were asked?

The measure of “experienced togetherness” in the context of the mirror game was introduced by Noy et al (2015; *Front Hum Neurosci*, <https://doi.org/10.3389/fnhum.2015.00187>). In the rated trials, participants were prompted with the instruction “*please rate the experienced*

togetherness during the previous trial". These details, and the reference, have been added to the manuscript (Methods -> Subjective measures) as follows:

"In every fourth block, participants were asked to rate their experienced togetherness of the preceding trial, for each trial in that block (a total of 4 times/condition), based on previous mirror game experiments by Noy and colleagues (Noy et al., 2015). Specifically, participants were prompted with the instruction "*please rate the experienced togetherness during the previous trial*". Responses were given on a 5-point scale [1 = low, 5 = high] using a keyboard, and could not be seen by the partner. "

(2) The methods section describes that "Sphericity could be assumed for all comparisons (all $p > .10$)." Does this mean that the threshold for the sphericity test was set at 0.10, and no values were below this threshold? Or is this just saying that all sphericity test values were greater than 0.10? If the threshold was indeed 0.10, can the authors give some reference for this value, since it's quite different from the typical 0.05 (or lower) threshold? Otherwise, I would recommend clarifying these statements so that the thresholds and the actual values are clearly given.

We thank the reviewer for catching this, and we would like to apologize for this confusion. The threshold for all comparisons and tests were set at $p < .05$; however, only values up to $p < .10$ are spelled out. Values above $p = .10$ are marked as $p > .10$ (given that, often, these concern a larger number of comparisons). We have now clarified the threshold to remove the confusion, see Methods -> Data analysis:

"Alpha levels for all comparisons and tests for assumptions for ANOVA tests were set to $p < .05$ unless noted otherwise."

(3) Functional Connectivity: I'm not very familiar with this type of FC analysis, but I found the description related to clustering to be a bit unclear. I think providing an example here would be helpful for the reader to know what a cluster might be.

We would like to apologize for the confusion regarding the clustering in the connectivity analysis. In an attempt to make it more clear, an example has been added to the description, as well as a link to the paper that motivated the clustering rules (Methods -> Data analysis -> Functional connectivity):

"Connections were categorized as neighbours when both ends of the connection were among the neighbouring (or the same) electrodes, adapted from the neighbourhood rules previously developed by Dumas and colleagues for inter-brain analyses (Dumas et al., 2010). For example, the connection between C3 and Oz was considered a neighbour of the connection C1-O1, since electrodes at both ends of the connection are neighbours (C3 is a neighbour of C1 and Oz is a neighbour of O1); as well as of the connection C3-O1, since O1 is a neighbour of Oz, and C3 is shared in both connections. However, Cz and Oz (or Cz-O1) is not a neighbouring connection of C3-Oz, as Cz and C3 are not neighbours or the same electrode. Consequently, if the differences between conditions passed the cluster forming threshold for multiple neighbouring connections, these were grouped together in a cluster for subsequent permutation testing."

*(4) For the EEG Task Effects results, from which participant are these results? The actor, observer, or average across both? **It would be helpful to specify this, both in the methods and results sections.** It seems to me that the results all come from one person in the dyad (i.e., the one performing the action, and presumably just one of those in the interaction condition). If the data indeed come from just one participant in each dyad, was anything done with the other participant's (i.e., the observer's) data?*

The EEG effects (as well as derived movement properties) were based on those trials in which a participant was performing actions, either observed or in interaction. Given that roles changed for the observed action condition (participant A observes participant B, then B observes A), all participants performed trials as (observed) actors. Therefore, each participant produced data as an observed actor in the observation condition, as well as an interacting partner in the interaction condition. Consequently, no selection or averaging was required. The data collected from the *observer* during the observation/observed action condition is not included in this study, given the different theoretical background, which is more established in the field of action observation.

A statement explaining this procedure has now been added to the manuscript (Methods -> Data analyses):

” It should be noted that data from both participants from all pairs (unless excluded for other reasons as mentioned above) were included in all analyses. Given that, in the observed action condition, all participants acted both as an actor and as an observer respectively (i.e. across half of the trials, participant A observed participant B, while in the other half of the trials participant B observed participant A), each participant produced data as an observed actor in this condition, as well as an interacting partner in the interactive condition. Consequently, no selection or averaging over partners in dyads was required, and all participants were included as individual observations. No data from trials in which a participant acted as an observer were included in the analyses presented here.”

(5) The mu suppression results are discussed as an index of integration, but there are many studies suggesting that alpha/mu suppression indexes attention, cognitive processing demand, or something similar. Could this also be an explanation for the findings? For example, the interactive task is likely much more challenging than the other conditions given the extra need to coordinate. The observed action condition may provide more pressure to perform “interesting” movements. In both cases, one could argue that there is no need to represent the other person in the dyad, only an increased task difficulty due to “better” (more interesting) performance, or producing movement that is easier to track (when leading) or otherwise to synchronize one’s own movement with what is being observed (when following).

We would like to thank the reviewer for these suggestions for alternative explanations. We have extended our discussion of “interpretational issues” to include and discuss these possibilities.

”Similarly, the observed mu suppression effects could also be related to attention or cognitive demand (Klimesch et al., 2007), mechanisms related to reduction in power of alpha oscillations over occipital areas (Van Diepen et al., 2015). Such attention/demand related suppression of alpha oscillations may therefore be related to the increased demands on coordination of one’s own movements with those of the partner in the interactive condition, as well as a perceived need to produce “better” movements (e.g. more interesting and creative, or less sluggish) in the observed action condition. As such, these

conditions would not necessarily require an integration of the partner or observer, but may simply be more challenging, either explicitly (due to being instructed to synchronize) or implicitly (due to a perceived pressure to produce “better” movements).”

We have also incorporated more relevant references to the function of alpha oscillations more broadly:

“More broadly, it has previously been suggested that alpha oscillations have a functional role related to attention and prediction (Klimesch et al., 2007), with the mu-rhythm reflecting a common coding mechanism of perception and action (Hari, 2006; de Lange et al., 2008). With regard to the present study, increased mu suppression during the interactive condition may thus reflect increased attention to self-other coupling of action-perception loops (Konvalinka et al., 2014), reflecting the integration of one’s own movements with those of the partner in order to achieve and sustain synchronized movements.”

(6) When discussing the intra-personal coupling results, the authors refer to Heggli and colleagues’ model of self-other integration as an explanation for the increased intrapersonal connectivity observed in the present study. Can the authors elaborate a bit on how intra-personal neural coupling relates to the perception-action coupling in the Heggli model? Particularly since the EEG results are quite general (i.e., not about a specific region-to-region connection).

We thank the reviewer for bringing up this important point. We have now made an effort to better explain the relationship to the Heggli study – as our hypothesis was that we would find more localized differences in functional connectivity. We also stress that the strongest connections are between fronto- and centro-parietal regions. Finally, we raise the concern that what we find here are indeed widespread differences in functional connectivity, which could be related to other task related differences, e.g. intensification of movement amplitudes.

A deeper link to Heggli et al. in the introduction:

“While coupled interaction increases inter-personal dependency (Dumas et al., 2010; Pérez et al., 2017), recent theoretical work suggests that performance of independent actions in interaction with others requires increased self-monitoring, which may be characterized by increased internal action-perception coupling during individual actions in a social context (Heggli et al., 2019). Conversely, interpersonal coordination of actions requires mutual prediction and adaptation, resulting in increased between-personal coupling of action-perception loops and decreased within-personal coupling (i.e. self-decoupling), presumably through a dynamic process of self-other integration (Heggli et al., 2019, 2021b). Previous work has shown that prefrontal, premotor, and parietal cortices are increasingly involved when participants attend to their own actions, as well as when they lead interactions (Fairhurst et al., 2014; Konvalinka et al., 2014). Furthermore, attention to one’s own actions has shown increased effective connectivity between prefrontal and premotor regions (Rowe et al., 2002). Following from this, we speculate that increased monitoring of one’s own actions in a social context is supported by higher fronto-parietal and prefrontal-premotor intra-brain connectivity, while self-other integration in joint actions is supported by decreased intra-brain connectivity – the latter representing evidence of self-decoupling during self-other integration on a neural level.”

and

“Moreover, we expect increased mu suppression in the observed actor, relative to an unobserved one, which would reflect either that the observer, despite not being overtly active, is represented in the observed actor’s motor system; or increased attention to the task, in this case, attention to self-produced actions.”

Interpretation of the results in the discussion:

“We observed a decrease in functional connectivity during interactions relative to individual actions (in isolation) with corresponding clusters of connections between occipital, parietal and temporal areas in the alpha frequency range, as well as increased widespread functional connectivity during observed actions. Notably, the strongest increases in functional connectivity (during observed actions) were found between frontal and centro-parietal regions (Figure 7b). Previous studies have shown connectivity changes between similar regions when participants attended to their own actions (Rowe et al., 2002), and functional connectivity in the alpha band has previously been tied to functions of cognitive control (Sadaghiani and Kleinschmidt, 2016). The prefrontal cortex is also known to be involved in higher level cognitive control of actions (Ridderinkhof et al., 2004; Koechlin and Summerfield, 2007), suggesting an increased level of self-monitoring and, possibly, higher attention towards own actions in the observed actor.

These results thus suggest that individual actions that do not rely on integration of actions of the other result in stronger intra-individual coupling, presumably due to an increase in monitoring of own actions. Interactions, on the other hand, require a tight coupling of self and other, hence a stronger inter-personal coupling, which we speculate may result in weaker intra-individual coupling due to reduced monitoring (and perhaps regulation) of own actions. This mechanism has recently been suggested following results from a Kuramoto model of self-other integration applied to interpersonal finger tapping data (Heggli et al., 2019). Heggli and colleagues showed that interpersonal coordination strategies rely on the dynamic balance between within- and between-person coupling of action-perception units. Namely, mutual adaptation (interaction) exhibited higher interpersonal coupling and lower intra-personal coupling of action-perception units. This was in contrast to “leader-leader” behaviour, where two people interact but do not adapt to one another – which was theoretically characterized by lower interpersonal coupling, and higher intra-personal coupling. While this is in line with the functional connectivity results in the present study, it should be noted that the functional connectivity differences here are generally widespread, instead of localized to fronto-parietal and prefrontal-premotor connections as hypothesized, which suggests intra-personal coupling effects that extend beyond action perception loops. We thus speculate that the widespread connectivity differences we see between the conditions may also be related to other task related differences, e.g. intensification of movement amplitudes.”

(7) More generally, given that you have the dual-EEG data, I was wondering why there were no analyses of inter-brain coupling, particularly since inter-brain results are brought up in the discussion, it seems this would be quite relevant.

We agree with the reviewer that the inter-brain coupling analyses are of high interest. However, given the different theoretical background (intra-brain vs inter-brain processes) and the huge differences in methodology we decided against inclusion of the inter-brain analyses

at this stage. However, we are working on an analysis of inter-brain processes based on the same dataset, which will be written up for another manuscript.

Minor:

(8) In the introduction, the authors provide a good background and motivation for the study, but I noticed that the studies on audience effects and communicative intentions in action kinematics were from 2011 and before, whereas there are a few more recent papers that I think would make a good addition to the literature review here.

For instance, Krishnan-Barman & Hamilton, 2019

(<https://doi.org/10.1016/j.cognition.2019.03.007>), McEllin et al., 2018

(<http://dx.doi.org/10.1037/xhp0000505>), Trujillo et al., 2018

(<https://doi.org/10.1016/j.cognition.2018.04.003>), Winner et al., 2019 (DOI: 10.1111/cogs.12733)

We would like to thank the author for her/his suggestions regarding additional literature. We have included the papers in the introduction of our manuscript (2nd paragraph).

“Increased movement amplitudes have also been observed in demonstrators of xylophone players when watched by a learner (McEllin et al., 2017), and in communicative actions (Trujillo et al., 2018). Using communicative pointing, it was shown that motor plans of communicative, observed actions are tuned to the addressee, by increasing the informational value of movements when seen from the addressee’s point of view (Winner et al., 2019). Audience effects in actions are not restricted to demonstrators, but also to followers. Specifically, followers imitate demonstrated movements more accurately when they are watched by the demonstrator (Krishnan-Barman and Hamilton, 2019).”

Reviewer: 2

#1 (minor)

The authors talk about the audience effect to introduce their research question. I’m wondering how different is the concept of audience effect from the social facilitation effect? As far I understand these two are really similar, and social facilitation has been studied in many different behavioral tasks, where the mere presence of the co-actor influences the behavior. What are the differences between audience effect and social facilitation? I believe that answering these questions would improve the theoretical framing of the presented experiment.

We would like to thank the reviewer for bringing up this question. Whereas audience effects and social facilitation are very similar, the difference is that audience effects require the observed person to know (or at least believe) that he/she is being observed by another person, whereas in social facilitation, the sheer presence of another person is sufficient to elicit an effect. This is now specified in the very beginning of the introduction to make it clear to the reader what we are aiming at.

“Audience effects can be distinguished from social facilitation, which are effects merely due to the presence of another person, but do not require the observed person to know – or believe – that he or she is actually being observed by this conspecific (Zajonc and Sales, 1966; Cottrell et al., 1968).”

#2 (minor)

The manuscript focuses on action monitoring and its neural underpinnings (measured with EEG), therefore, I would like to suggest that a classic paper showing the effect of observation during action monitoring should be referenced:

van Schie, H., Mars, R., Coles, M. et al. Modulation of activity in medial frontal and motor cortices during error observation. Nat Neurosci 7, 549–554 (2004).

<https://doi.org/10.1038/nn1239>

We would like to thank the reviewer for suggesting this classic and high quality paper on the neural underpinnings of action observation and error observation in particular. We have now incorporated this reference in the first paragraph, where we refer to studies of observation.

#3 (minor)

The authors report that they used two different groups of dyads (strangers and familiar (friends and couples)). There is much evidence that joint actions/social interactions are heavily affected by the relations between members of the dyad. I'm wondering why authors decided to mix different types of dyads? Maybe it should be additionally analyzed whether effects reported in the manuscript are the same for both groups (strangers vs. familiar).

The initial decision to mix different types of dyads was done for practical reasons during participant recruitment (i.e. participants could sign up alone or at the same time as a familiar person, and we did not have to make sure in advance whether participants know each other, either because they signed up together, or by accident, as all participants were recruited through the university network).

Given low expected power with regard to analyses comparing effects between dyad types, we do not include those analyses in the main manuscript. However, we now compared the movement and EEG ROI data between groups (friends vs non-friends) using two sample t-tests. None of the comparisons returned a significant effect. This does not mean there are no differences, as the study, and the number of pairs per group, is not designed for such a comparison. These results are added to the supplementary material (S5) and notes were added to the appropriate sections in the results for movement properties and EEG effects.

“No significant differences have been observed between dyads formed by friends and dyads formed by strangers (Supplementary material S5).”